

# 1 Frequency of large volcanic eruptions over the past 200,000 years

Eric W. Wolff[1], Andrea Burke[2], Laura Crick[2], Emily A. Doyle[1], Helen M. Innes[2], Sue H. Mahony[3], James
W.B. Rae[2], Mirko Severi[4] and R. Stephen J. Sparks[3]
[1]Department of Earth Sciences, University of Cambridge, Cambridge CB2 3EQ, United Kingdom
[2]School of Earth & Environmental Sciences, University of St Andrews, St Andrews, KY16 9AL, United Kingdom
[3]School of Earth Sciences, University of Bristol, Bristol BS8 1RJ, United Kingdom
[4]Chemistry Department, University of Florence, Sesto F.no (FI) 50019, Italy
*Correspondence to*: Eric W. Wolff (ew428@cam.ac.uk)
**Abstract.** Volcanic eruptions are the dominant cause of natural variability in climate forcing on timescales up to multidecadal.
Large volcanic eruptions lead to global-scale climate effects and influence the carbon cycle on long timescales. However,
estimating the frequency of eruptions is challenging. Here we assess the frequency at which eruptions with particular deposition
fluxes are observed in the EPICA Dome C ice core over the last 200 kyr. Using S isotope analysis we confirm that most of the
largest peaks recorded at Dome C are from stratospheric eruptions. The cumulative frequency through 200 kyr is close to linear
suggesting an approximately constant rate of eruptions. There is no evidence for an increase in the rate of events recorded in
Antarctica at either of the last two deglaciations. Millennial variability is at the level expected from recording small numbers
of eruptions, while multimillennial variability may be partly due to changes in transport efficiency through the Brewer-Dobson
circulation. Our record of events with sulfate deposition rates $> 20$ mg m$^{-2}$ and $>50$ mg m$^{-2}$ contains 678 and 75 eruptions
respectively over the last 200 kyr. Calibration with data on historic eruptions and analysis of a global Quaternary dataset of
terrestrial eruptions indicates that sulfate peaks with deposition rates $> 20$ mg m$^{-2}$ and $>50$ mg m$^{-2}$ correspond to explosive
eruptions of magnitude $\geq 6.5$ and $\geq 7$ respectively. The largest recorded eruption deposited just over 300 mg m$^{-2}$.

## 22 1. Introduction

Volcanic eruptions can have devastating local effects, and at a global scale they are one of the important natural components
of forcing in the climate system (Robock, 2000). The forcing arises from sulfate aerosol that is formed from $SO_2$ erupted into
the stratosphere. On longer timescales the balance between volcanism and weathering controls the $CO_2$ content of the
atmosphere, and changes in volcanic eruption frequency could contribute to the changes in $CO_2$ concentration observed at
glacial terminations (Huybers and Langmuir, 2009). In order to constrain changes in forcing by volcanic aerosol, as well as
any role of volcanoes in glacial-interglacial $CO_2$ change, a key question is whether global eruption rate is steady and, if not,
whether any variation is related to climate.
There has been much interest in the notion that rates of explosive volcanism have been moderated by processes related to
climate change (Kutterolf et al., 2019; Watt et al., 2013). Rates of mantle melting are expected to be affected by glacial cycles:



melting of ice caps leads to unloading, enhanced mantle melting and enhanced volcanism (Huybers and Langmuir, 2009; Jull
and McKenzie, 1996). Rates of volcanism might also be affected by sea level change (Huybers and Langmuir, 2009; Kutterolf
et al., 2019). Over very long timescales, changes in plate tectonics and occurrence of mantle plume volcanism are expected to
be reflected in rates of volcanism.
Beyond the period of direct historic observations, explosive eruptions are recorded as tephra deposits in terrestrial and marine
records, and as both sulfate and occasional tephra deposits in ice cores. Terrestrial tephra deposits give information about
location as well as strength and frequency of eruptions (Brown et al., 2014), but they are notoriously difficult to use, and a
statistical approach is needed to turn them into useful measures of eruption frequency (Rougier et al., 2018). Tephra in marine
cores offers a further opportunity to compile eruption statistics (Mahony et al., 2020), but it is also challenging to compile a
record that is unbiased over time and space. Eruptions are also recorded as sulfate deposition in ice cores. While this provides
no direct information on the location and magnitude of each eruption, it can be used to log eruptions relevant for climate
forcing (Gao et al., 2008; Sigl et al., 2015). There have been only limited investigations (Castellano et al., 2005; Castellano et
al., 2004; Cole-Dai et al., 2021; Lin et al., 2022) to estimate volcanic eruption occurrence from ice core data beyond the last
2500 years.
Although records from both poles may be combined to identify explosive eruptions recorded at both poles (and therefore most
likely having reached the stratosphere), the use of Greenland ice core records alone is complicated, because they are dominated
by the relatively local input from Icelandic eruptions. Antarctica has rather few local eruptions (notwithstanding an unusual
event at the last deglaciation (McConnell et al., 2017)), and so the record of eruptions is likely dominated by those that have
reached the stratosphere and have a global climate effect.
Eruptions can be logged using spikes in sulfate, or as a surrogate, spikes in the dielectric profile conductance (DEP) or low
frequency electrical conductivity (ECM) of ice (Wolff, 2000), both of which respond to acidity in the ice. Such records are
available continuously from a number of ice cores on the East Antarctic plateau. Issues of resolution and diffusion mean that
the volcanic spikes cannot be reliably observed to the bottom of the cores, but they are clearly identified over at least the last
two glacial cycles. This has for example allowed their use to synchronise age models to 128 ka bp between EPICA cores at
Dome C (EDC) and Dronning Maud Land (EDML) (Ruth et al., 2007), to 145 ka between EDC and Vostok (Parrenin et al.,
2012), and to 216 ka between EDC and Dome F (Fujita et al., 2015).
Several issues make it challenging to construct a consistent record of sulfate deposition throughout an ice core. Firstly, there
is a background of sulfate from non-volcanic sources (mainly sea salt and marine biogenic). Because this background, and its
variability, changes with climate, methods which merely seek outliers (Castellano et al., 2004) risk recording volcanic events
depositing a particular amount of sulfate in some climate periods and not in others. Secondly there is a very large amount of
variability in the amount of sulfate deposited in the small footprint of snow surface sampled by an ice core (Gautier et al.,
2016; Wolff et al., 2005), so that the sulfate signal of individual eruptions in a single core is subject to great uncertainty.



Thirdly, Antarctic ice cores will record some eruptions which did not reach the stratosphere but are smaller eruptions of more
regional origin. These can in principle be filtered using sulfur isotope analysis to identify mass independent fractionation
(Baroni et al., 2008; Burke et al., 2019; Gautier et al., 2019; McConnell et al., 2017; Savarino et al., 2003a).
A final issue is related to resolution and diffusion. Snow accumulation rates at the East Antarctic sites mentioned above (Dome
Fuji, EDC, Vostok and EDML) vary from 2-6 cm water equivalent at the different sites in the present day and are typically
less than 50% of that in the last glacial maximum. Eruption signals are generally recorded above background for only 2-3
years. It is therefore essential to use only data with a good depth resolution, and to estimate the flux across the peak and not
just at the maximum, which will certainly be modulated by the resolution. Additionally sulfate peaks diffuse with age (Barnes
et al., 2003), and we find that volcanic peaks that were just a few years wide on deposition may appear to be 20 years wide in
ice of 200 ka age at EDC. This is helpful because it means that, as layers thin with depth, the decreasing age resolution of our
measurements is not a limiting factor. However it makes it yet more challenging to identify eruptions of a particular scale in a
consistent way, because peaks that stand clearly above background in recent ice diffuse towards the background in older ice.
It is also important to be aware that even a perfect record of sulfate deposition events is not easy to convert into a record of
eruption magnitude and frequency. In addition to the strength of the eruption, sulfate deposition depends on the sulfur content
of the eruption, the latitude of the volcano and atmospheric transport processes (Marshall et al., 2021). Finally, there are
difficulties in converting sulfate loading in an ice core to the magnitude (defined by volcanologists as magma mass erupted).
Eruption magnitude is only one of several factors that influence sulfate mass released in explosive eruptions (Wallace and
Edmonds, 2011).
Despite these problems two recent papers (Cole-Dai et al., 2021; Lin et al., 2022) have attempted to assess eruption frequency.
One (Cole-Dai et al., 2021) examined the number of eruptions recorded in the WAIS Divide core over the last 11 kyr, observing
variability but no trend in eruption rate. A second (Lin et al., 2022) assessed eruption frequencies recorded in both Greenland
and Antarctica over the period from 9-60 kyr ago, and compared them with the rates in the last 2 kyr. For Greenland they
found relatively constant rates of eruption through time, but with a small increase in frequency of recorded eruptions across
the deglaciation (21-9 ka). This is consistent with the idea that the removal of ice from high latitude eruption regions, with a
particular emphasis on Iceland (Jull and McKenzie, 1996), would have led to an increased eruption rate recorded in Greenland.
For Antarctica, using methods similar to those we describe later, the authors (Lin et al., 2022) found no significant change in
eruption rate across the 60 kyr period.
In this paper we log eruption frequencies from the Antarctic ice core of EDC to 200 ka, using a methodology that assesses the
scale of sulfate deposition consistently with depth, age and climate period.



## 2. Data

Sulfate was measured along the EPICA Dome C (EDC) ice core by two methods, standard ion chromatography (IC) and fast ion chromatography (FIC) (Littot et al., 2002; Severi et al., 2015). FIC measurements were at higher resolution (typically 5-6 cm in the top 100 m, 3-5 cm below that to 770 m, and 2 cm below 770 m) than IC measurements; additionally there are sections of the core where IC data are not available. For these reasons FIC data were preferred.

However, although initial tests had suggested good agreement between the methods (Littot et al., 2002), a more detailed analysis suggested some calibration problems during the first field season of FIC use, to a depth of 358.6 metres (11745 years before present (1950) on the AICC2012 age model). In this depth range we consider the well-established IC method to be more reliable. Detailed comparison of FIC and IC data was carried out where both were available. While it is impossible to diagnose exactly what the issue was, a plot of FIC data against IC data for values more than 50 µg kg$^{-1}$ above the background showed a gradient of 0.70 for data between 0 and 358.6 m, and 0.94 between 360 m and 720 m. This is shown in Fig. 1 and was used as justification to multiply the FIC values above background (i.e. the residual after subtracting the background) by (1/0.7) for data above 358.6 m (<11.7 ka) during the data processing.





**Figure 1.** Comparison of FIC and IC data for two sections of the EDC ice core. In the section from 130 to 145 m, FIC peaks are consistently lower than those of IC while in the section from 390 metres, the concentrations are essentially the same in the two methods.

Beyond 358.6 m, the data were used without correction. Volcanic peaks, standing clear of the sulfate background, remain visible beyond 200 ka. It is clear that diffusion has occurred, making peaks considerably wider in years (and consequently smaller in amplitude) than they were at the time of deposition; however, thinning seems to balance diffusion rather closely (Fig. 2). The result is that peaks remain within about a 30 cm window at all depths to 200 ka, and the resolution of the data remains adequate to estimate peak areas.

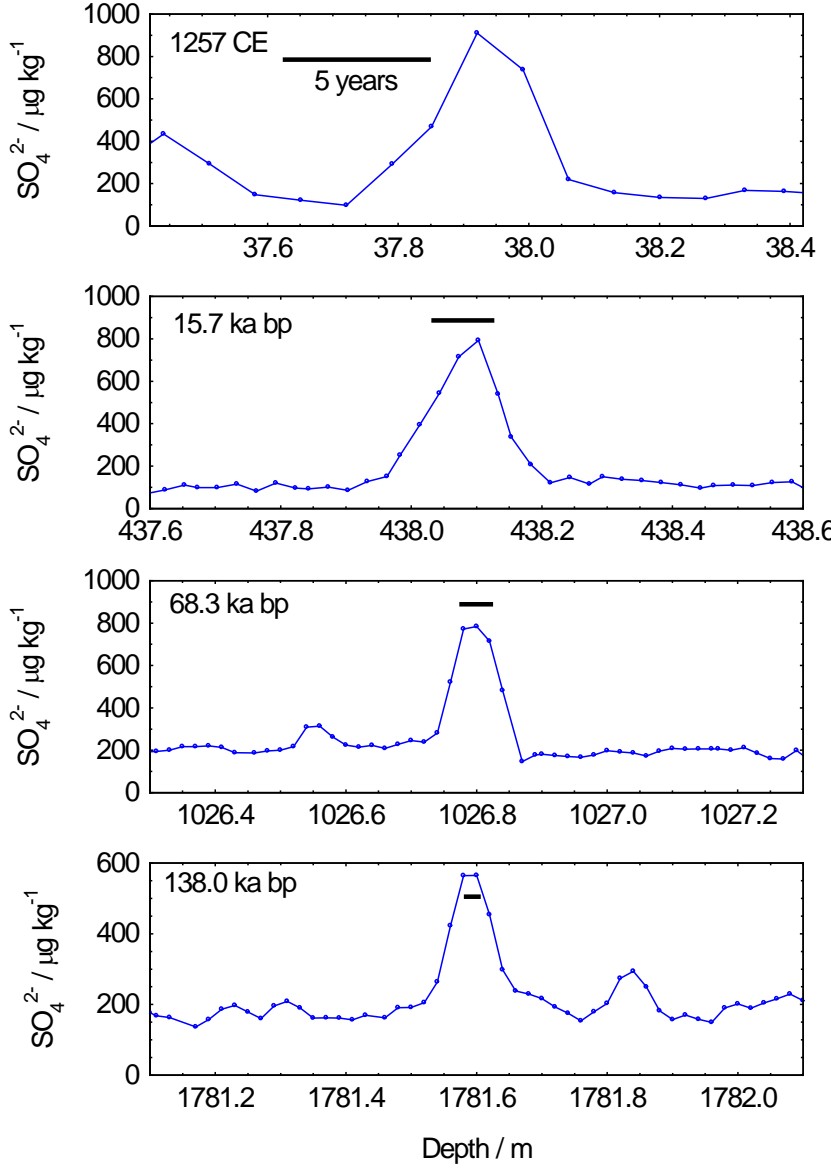

Figure 2. Examples of volcanic peaks at different depths and ages. The 1257 peak (top panel) is shown after application of the correction described above. The black horizontal bar on each plot represents 5 years at each depth. Dots represent the mid-depths of individual samples.

The dataset includes sections of missing data, where no FIC sulfate data are available. Mainly this consists of short sections at the end of core lengths, but there are some longer sections where data were not taken either because of poor core quality or instrument problems. Out of 2070 m of ice, there are 25 gaps longer than 30 cm, consisting in total of 19 m of ice. We discuss our treatment of missing data under methods.





In deeper ice, it has been observed (Traversi et al., 2009) that anomalous spikes in sulfate concentration form through an as-
yet uncharacterised post-depositional process. In Fig. 3 we show clear examples of this artefact at about 400 ka; sulfate appears
to have been "sucked" from the surrounding background into a sharp peak. We observed signs of this behaviour as shallow as
2500 m (300 ka). To avoid any possibility of including such artefact peaks, we restrict our subsequent analysis to the past 200
kyr. This also avoids the problem that peaks become harder to distinguish from background with greater depth.

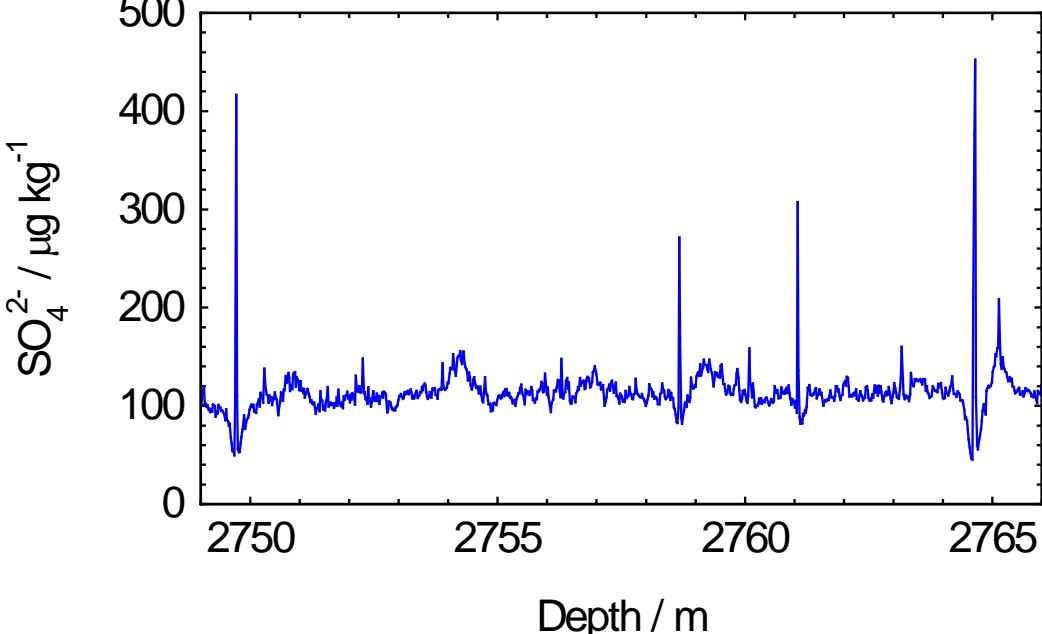


Figure 3. An example of four artefact peaks in ice aged just over 400 ka.
Material for the last few decades was not available in the EDC ice core because the top few metres were not retrieved. However,
the Pinatubo period has been studied previously at Dome C using snow pits (Castellano et al., 2005). The observed peak
(deposition of 10.7 mg m$^{-2}$) encompassed both Pinatubo and the eruption of Mount Hudson in Chile, which could not be
separated.  However, the two eruptions have been resolved at South Pole (Cole-Dai and Mosley-Thompson, 1999), allowing
a fraction of deposition in the combined peak to be assigned to each eruption. Using the same fraction, we estimate 7.5 mg m$^{-}$
$^{2}$ for the Pinatubo sulfate deposition at Dome C, used later as part of benchmarking our data.
**3.  Methods**
We applied the following method of calculating sulfate deposition. The ice core volcanic record consists of numerous sharp
spikes of sulfuric acid input, superimposed on a noisy background. The background consists mainly of sulfate from oxidation



of marine biogenic emissions of dimethylsulfide, with small contributions from sea salt as well as background volcanic sulfate.
Based on sulfate concentration measurements (Legrand and Delmas, 1984) and measurements of $\delta^{34}S$ in ice at South Pole
(Patris et al., 2000), the volcanic contribution to the background is estimated as less than 10%. In order to calculate the sulfate
deposition during each individual eruption event, we subtract the background and then sum the area across the peak, correcting
for ice thinning.
To correct for the background, we subtracted a running median from the dataset. The median is preferred to the mean because
the mean includes the volcanic peaks while the median should, if well-chosen, exclude the peaks. The time period over which
the median is calculated needs to be short enough that it follows the varying background but long enough that it will never use
the values within volcanic peaks. In our standard calculations we used 200 years, but other periods were also tested in
sensitivity studies. Varying the period over which the median was calculated between 100 and 400 years changed the total
number of peaks above a threshold of 20 mg m$^{-2}$ by up to 10% compared to the standard case (200 years) but did not affect the
profile of peaks with time.
Having subtracted the background, we calculate the total amount of sulfate deposited to the snow per unit area across the whole
eruption. A key assumption is that the sulfate is mainly deposited by dry deposition, which is expected to be true at a site like
Dome C with its very low snow accumulation rate. This justifies the underlying assumption that the sulfate flux scales with
the amount of sulfate injected into the stratosphere. We first calculate the annual flux of sulfate in each sample ($\mu g$ m$^{-2}$ a$^{-1}$) as
$F = C * A$, where $C$ is the concentration in a slice of ice ($\mu g$ kg$^{-1}$), and $A$ is the snow accumulation rate (kg m$^{-2}$ a$^{-1}$). We then
calculate the total deposition in each sample ($\mu g$ m$^{-2}$) by multiplying the flux by the time period represented in each sample;
this is done by using the accumulation rate and thinning parameter derived in the AICC2012 age model (Bazin et al., 2013) to
calculate the annual layer thickness. Finally, since the FIC data are effectively averages for discrete slices, we sum the
deposition (above background) in each of the samples that contribute to a particular peak to get the total deposition for an
individual eruption. This method automatically corrects the flux for the ice thinning (which is already 73% at the depth (2090
m) of 200 ka ice.
The algorithm we use searches for local maxima in the residual (after subtraction of the background) and calculates the sum
of samples across a chosen summing width across each maximum. The summing width needs to be large enough to include
all the volcanic sulfate after diffusion (Barnes et al., 2003). Visual observation suggests that a width of 30 cm (ie samples
within 15 cm of the concentration maximum) is appropriate at all depths between the surface and 2100 m. This width is justified
because diffusion more or less keeps pace with thinning at EDC. However, this width was also varied in sensitivity studies.
Varying the integration width between 20 and 40 cm altered the total number of peaks above a threshold of 20 mg m$^{-2}$ by up
to 10% compared to the standard case (30 cm) but did not affect the profile of peaks with time. At many depths, an integration
width of 20 cm is clearly too narrow to capture the full peak, while 40 cm includes sections of background, so the uncertainty
induced by this parameter is below 10%.



Our method calculates numerous small peaks that are caused simply by variations around the background. To estimate this
variation we also calculate "negative" peaks around our median line. We then separately sum the number of peaks and negative
peaks in bins exceeding particular deposition fluxes (Figure 4). At a deposition of 10 µg m$^{-2}$, there is still a substantial number
of negative peaks (441 in 200 kyr, compared to 1518 positive peaks). At 20 mg m$^{-2}$, there are very few negative peaks (28 in
200 kyr compared to 678 positive peaks), suggesting that 96% of peaks we count at this level are volcanic eruption peaks, and
supporting our choice of 20 mg m$^{-2}$ as the background threshold for counting peaks. There are no negative peaks at 40 mg m$^-$
$^2$.  This indicates that, while we could investigate volcanoes with lower deposition fluxes in some time periods, we should
restrict ourselves to peaks above 20 mg m$^{-2}$ in order to count peaks of similar size consistently over 200 kyr.
In our standard calculation we treated missing data as having the concentration of the background, i.e. those sections did not
contribute to the size of volcanic peaks in which they were embedded. We also did a calculation where we set the value of all
missing sections less than 30 cm thick to be the average of the adjacent samples: this increased the total count of peaks > 20
mg m$^{-2}$ by only 11 (out of 678). It is likely that the longer sections of missing data (25 sections > 30 cm, totalling 19 m of ice)
would have contained some peaks but assuming they contain the same proportion of volcanoes as the measured parts we have
probably missed less than 10 peaks with deposition >20 mg m$^{-2}$.

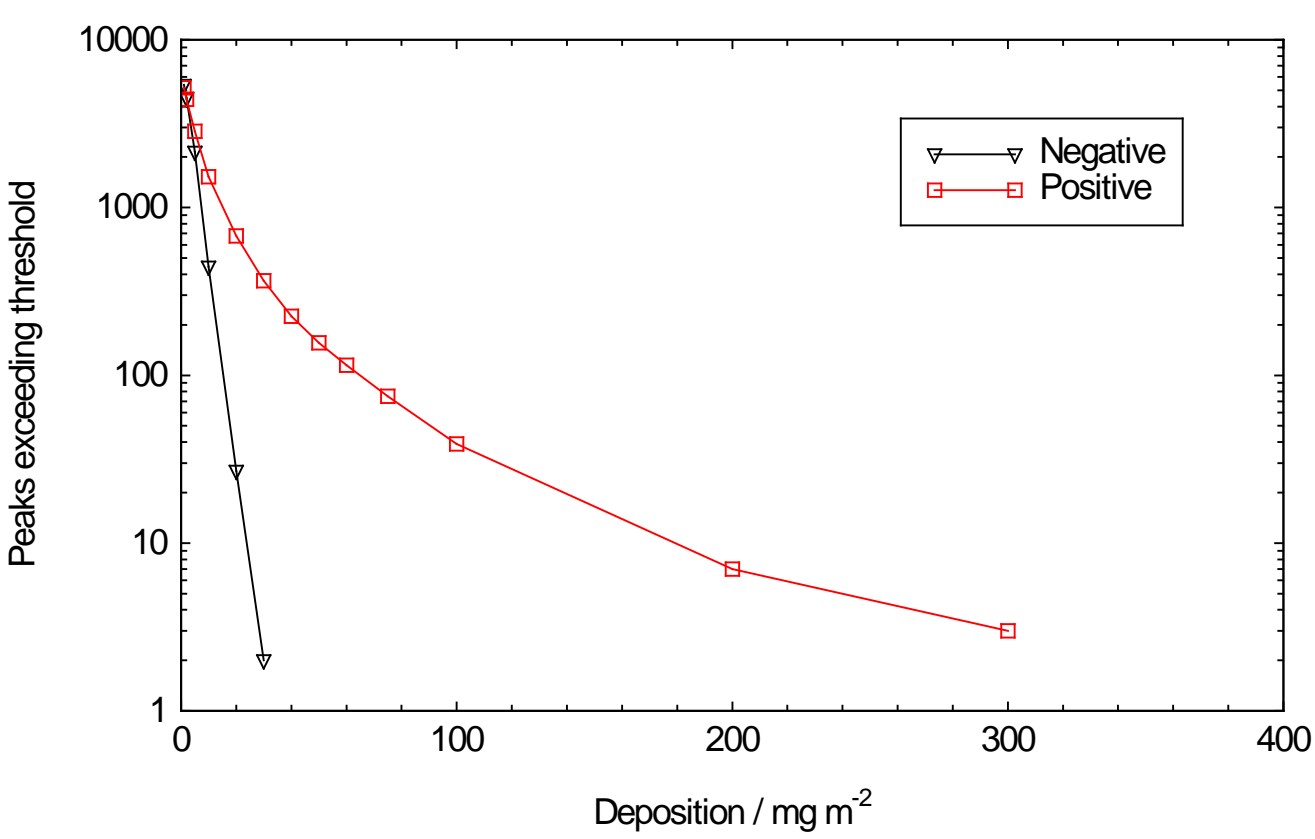






Figure 4. Distribution of positive and negative peaks exceeding different deposition fluxes, summed over the past 200 kyr.
Sulfur isotopes were measured on discrete samples of ice cut at high resolution (every 2-3 cm) across 21 volcanic sulfate
events from Dome C between 10.1 and 96.1 ka. Samples were melted and measured for concentration by ion chromatography.
Based on the concentration, a volume corresponding to 20 nmol of sulfate was dried down and purified through anion exchange
columns following the method previously described (Burke et al., 2019). Each sample was measured at least twice for $\delta^{34}S$
and $\delta^{33}S$ by multi-collector inductively coupled plasma mass spectrometry, where
$\delta^{x}S = (^{x}S/^{32}S)_{sample} / (^{x}S/^{32}S)_{reference} - 1$
and x is either 33 or 34. Mass independent fractionation was calculated as
$\Delta^{33}S = \delta^{33}S - ((\delta^{34}S + 1)*0.515 - 1)$
The uncertainty for these $\Delta^{33}S$ measurements is 0.14‰ (2 s.d.). Only sulfate that has been in the stratosphere shows a non-
zero signal of $\Delta^{33}S$, and so if the maximum magnitude $\Delta^{33}S$ across a peak is greater than 0.14‰, the eruption is considered
stratospheric.

## 4.  The frequency of eruptions recorded in Antarctica

A few of the most recent sulfate layers can be correlated to specific eruptions allowing some calibration of the record to the
magnitude of explosive eruptions (Gao et al., 2008; Sigl et al., 2015), but most layers cannot be linked to a source.  As a
benchmark Table 1 lists four sulfate peaks in Dome C where the eruption location and magnitude are also known, including
the 1991 Pinatubo eruptions (Castellano et al., 2005). The benchmark data (all from tropical eruptions) imply that peaks above
our chosen threshold of 20 mg m$^{-2}$ are likely to be M>6.5 eruptions.

| Eruption | Dome C deposition (mg m$^{-2}$) | Magnitude | Emission (Tg SO$_2$) |
|---|---|---|---|
| Pinatubo 1991 | 7.5 | 6.1 | 18 |
| Krakatoa 1883 | 13.2 | 6.4 | 19 |
| Rinjani/Samalas 1257 | 74.5 | 7.0 | 119 |
| Tambora 1815 | 53 | 7.0 | 56 |


Table 1. Identified sulfate peaks in Dome C with magnitude and estimated emission (Toohey and Sigl, 2017).
Various attempts have been made to derive SO$_2$ emissions (in Mt or Tg of S or SO$_2$) from ice core deposition (in mg m$^{-2}$ of
sulfate) (Sigl et al., 2022). However this is difficult when there is only one ice core location, and the location of the eruption
is unknown. Model studies show that the ratio of deposition in Antarctica to emissions depends on latitude of eruption, the
height the plume reaches, and the time of year of the eruption (Marshall et al., 2021). As a rough estimate using emissions
values calculated in the literature (Toohey and Sigl, 2017) we can deduce that for tropical eruptions, SO$_2$ emissions in Tg SO$_2$





are about 1-2 times higher than our measured EDC depositions in mg m$^{-2}$ (Table 1). However the factor should certainly be
increased if eruptions occurred at high northern latitudes (Marshall et al., 2021).
Using our base set of parameters (200 year median calculation, 30 cm summation of layers per volcano, missing values treated
as having background concentration), we find that the last 200 kyr contains 678 volcanic events with deposition rates greater
than 20 mg m$^{-2}$ (Fig. 4); this gives an average of 3.4 per millennium. Although our method is identical in concept, we calculate
rather more peaks greater than 20 mg m$^{-2}$ (2.87/ka vs 2.21/kyr) for the period 9-60 ka than that estimated for EDC by previous
work (Lin et al., 2022). This difference seems to arise because our method calculates higher integrals for smaller peaks,
suggesting that the difference is related to the way that the background is calculated and/or the way that we deal with the width
of each peak. This is supported by the fact that at the extreme of our parameter choices (20 cm peak widths, and 100 year
interval for calculating the median background) our estimates converge with those of the previous work (Lin et al., 2022).
There are only 76 peaks with fluxes larger than that of Rinjani/Samalas (1257), making this a 1 in 2500 year event. A time
series of all eruptions greater than 20 mg m$^{-2}$ is shown in Fig. 5.
Our results are also consistent with independent estimates of the global magnitude-frequency relationship (Rougier et al.,
2018). Based on data shown in Table 4, sulfate peaks > 20 mg m$^{-2}$ should have magnitudes ≥6.5 while sulfate peaks > 50 mg
m$^{-2}$ should have magnitudes ≥ 7. The analysis of global terrestrial data (Rougier et al., 2018) gives an estimate of M ≥ 6.5
eruptions as 2.75/kyr (confidence interval (CI) 1.6-4.3) and an estimate of M ≥ 7 eruptions as 0.8/kyr (CI 0.48 to 1.47). Thus
the event rates based on sulfate events at > 20 mg m$^{-2}$ (3.4/kyr) and > 50 mg m$^{-2}$ (0.78/kyr) are well within the uncertainty
ranges of the estimates from the global terrestrial record.
The largest peaks in the past 200 kyr deposited around 300 mg m$^{-2}$. The largest recorded eruption in the timeframe that could
accommodate the Toba eruption (Crick et al., 2021; Svensson et al., 2013) has a flux of 133 mg m$^{-2}$ (16[th] largest in our record),
Thus it unlikely that, in terms of global dispersion of sulfate aerosol, Toba was the most significant volcanic climate forcing
event of the past 200 kyr.



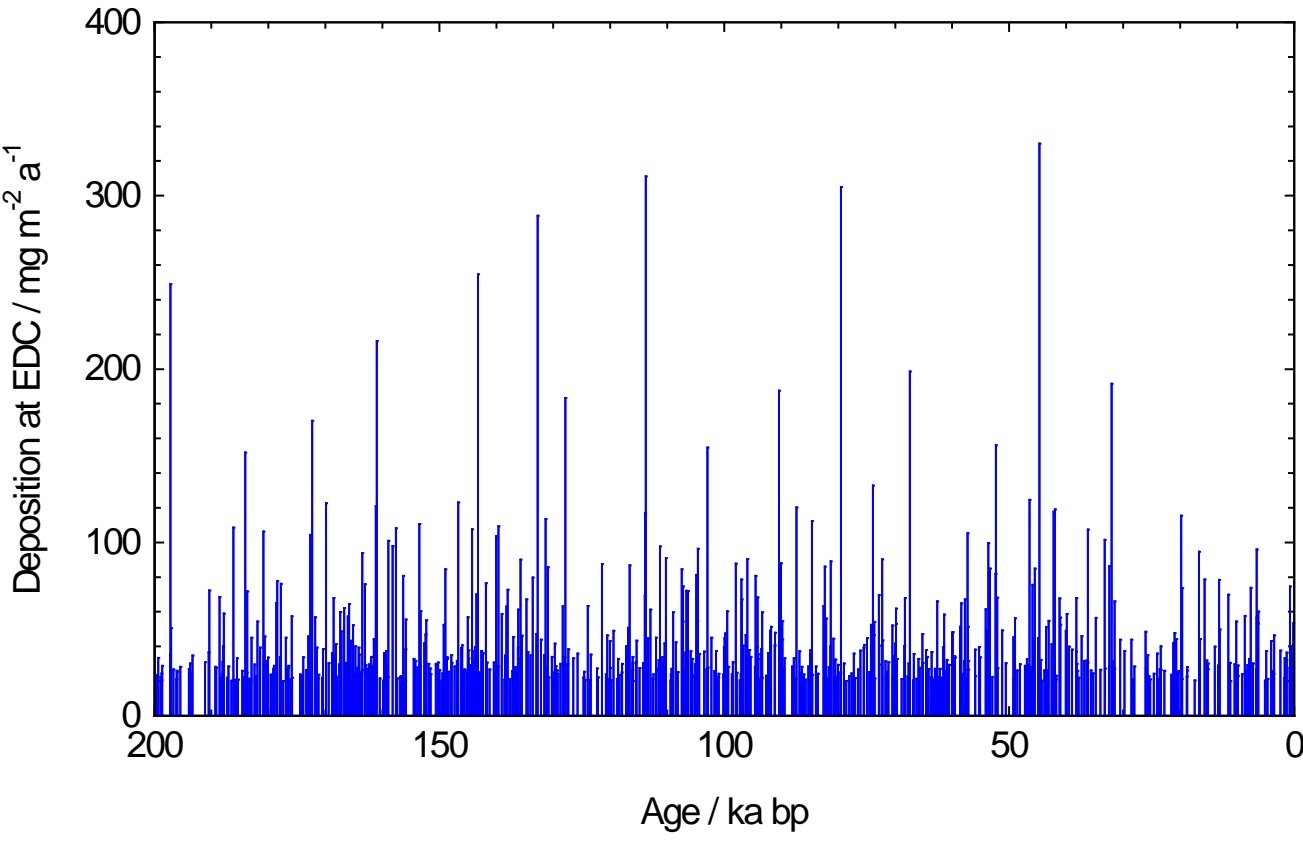

Figure 5. The deposition flux of sulfate for events with deposition more than 20 mg m$^{-2}$ over the last 200 kyr.

To assess whether there are particular periods with high or low numbers of eruptions, we plot the cumulative number of large sulfate deposition events with time (Figure 6). Concentrating mainly on the result for eruptions greater than 20 mg m$^{-2}$ because of the greater numbers involved, the trend is linear, indicating a steady state of large explosive eruptions across two glacial cycles. There is no sign of an increased slope (i.e., increased eruption frequency) at the two periods of deglaciation or interglacials. This can be seen in Fig. 7. The number of eruptions per millennium is very variable, as is to be expected from counting statistics for such small numbers. As a result the time series plot of the occurrence data is very scattered (Fig. 7). Nonetheless it is quite obvious that, at Dome C, both periods of deglaciation tend to have eruption frequencies at the lower end of the range, rather than increased rates.



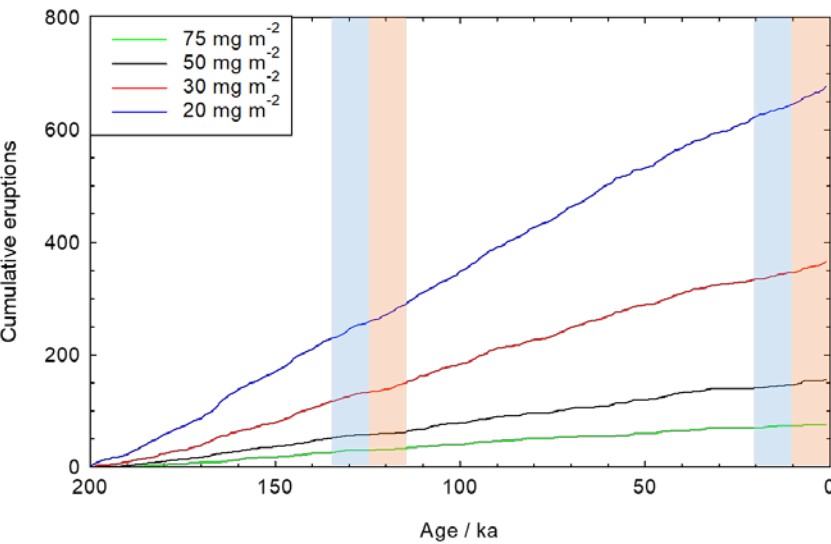

244

Figure 6: Cumulative eruption numbers over the past 200 kyr recorded in the Dome C ice core for different eruption deposition fluxes of sulfate. Periods of deglaciation marked with blue bar, interglacials with orange bar.

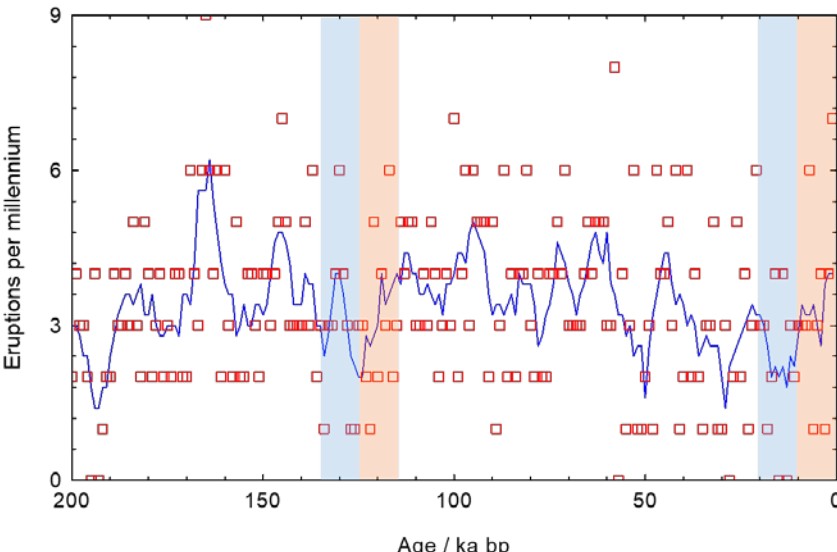

247

Figure 7: Eruption numbers per millennium (red) and 5 kyr running mean (blue). Periods of deglaciation marked with blue bar, interglacials with orange bar.

We explored other ways of analysing the ice core data to assess whether a climate cycle signal can be recognised. Using spectral analysis on the millennial eruption counts, we identified a possible peak corresponding to a 20 kyr period (frequency similar to precession) that emerges with weak statistical significance. This certainly needs to be confirmed in other records.





Although there have been weak indications of a 23 kyr period in Mediterranean tephra data (Kutterolf et al., 2019) it is difficult
to envisage a mechanism by which precession would influence global volcanism, given that it leads to much weaker changes
in ice sheet unloading and sea level compared to the longer (of order 100 kyr) period. Temperature and snow accumulation
rate at the EDC site show only very weak precessional power (Jouzel et al., 2007), so precessional changes in deposition
efficiency are unlikely to be strong. However, precession significantly influences tropical hydroclimate and the position and
width of the ITCZ (Singarayer et al., 2017). These changes could affect the washout of aerosol from eruptions in tropical
regions, and hence their ability to reach the stratosphere. There will certainly also be an associated effect on the efficiency of
the Brewer Dobson circulation that transports aerosol to the poles through the stratosphere, although we are not aware of model
simulations of this transport involving significant changes in precession (Fu et al., 2020). Thus, if the 20 kyr period is confirmed
it might be ascribed to a small change in the effectiveness of transporting tropical or Northern Hemisphere eruption material
to Antarctica. We emphasise that there is no significant signal at the lower Milankovitch frequency corresponding to 40 kyr.
Our record is too short to make a meaningful assessment in frequency space of the ~100 kyr cycle on which deglaciations
occur, but we again emphasise that, if anything, we see lower numbers of recorded events across the two deglaciations.
**5.  Discussion**
As described earlier, there are several challenges when interpreting this dataset as a record of global volcanism. First, there is
the difficulty of separating local tropospheric from larger magnitude stratospheric eruptions. Mass-independent sulfur isotopes
in ice cores can be used to determine whether a volcanic event was stratospheric (Baroni et al., 2008; Burke et al., 2019;
Gautier et al., 2019; Savarino et al., 2003b). Isotope analysis of large sulfate peaks from the last 2600 years at Dome C (Gautier
et al., 2019) indicates 11 tropospheric and 49 stratospheric events, with 4 events showing an inconclusive signal.  All the
largest events ($>20$ mg m$^{-2}$ deposited at Dome C) were stratospheric. In this study, we tested if the proportion of stratospheric
events recorded at Dome C was the same earlier in the record by measuring an additional 21 events from Dome C between
10.1 and 96.1 ka following previous methods (Burke et al., 2019). We found (Fig. 8 and supplementary table) that most (18
out of 21; 15 out of 17 for deposition $>20$ mg m$^{-2}$) volcanic signals in Dome C are stratospheric, consistent with its isolated
location away from most volcanic sources. The sulfur isotope data therefore show that more than 80% of the volcanic events
recorded at Dome C involved stratospheric input due to large explosive eruptions.





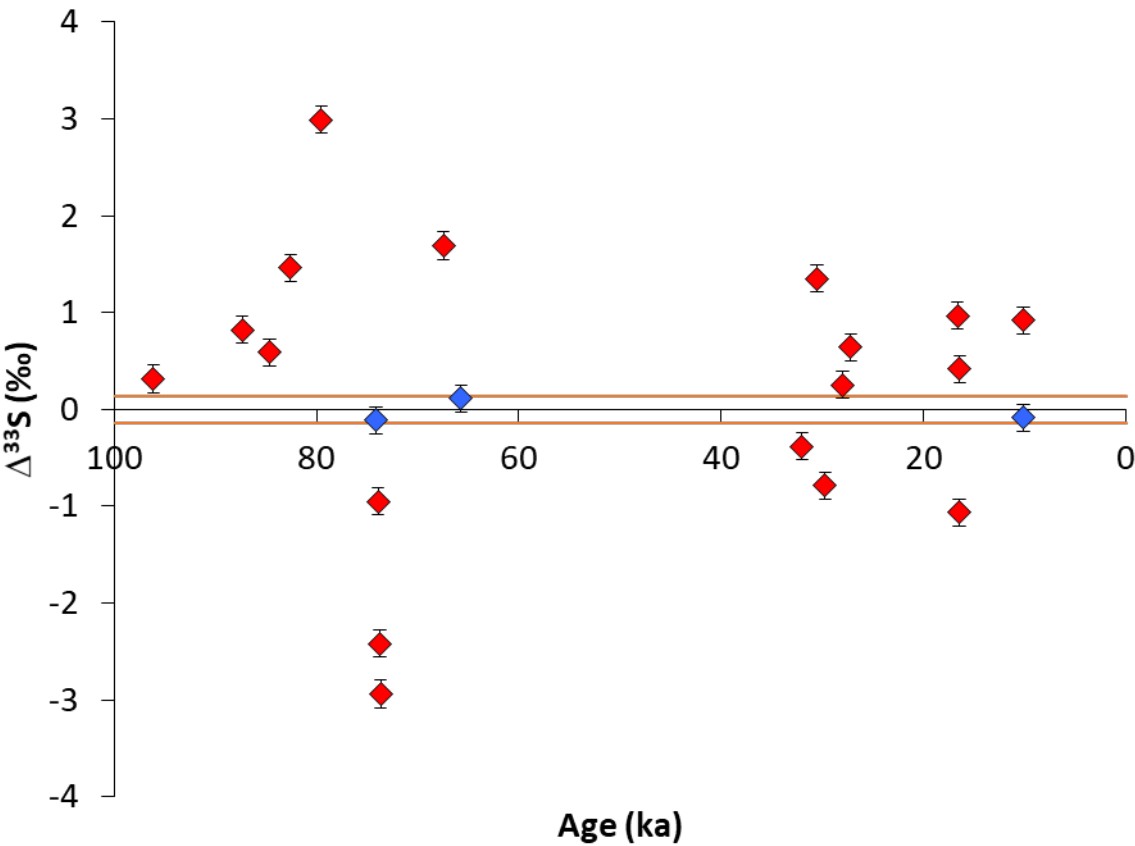


Fig. 8. Values of $\Delta^{33}S$ for 21 large volcanic eruptions recorded at Dome C in the last 100 kyr. Values outside a range of ±0.14‰ are considered to indicate a stratospheric eruption.

Second, sulfate peak amplitudes can vary strongly between core sites that are close together, and major peaks can even be missing at a single site (Gautier et al., 2016; Wolff et al., 2005). This problem is particularly pronounced at sites with low snow accumulation where years may be missing, like Dome C (Wolff et al., 2005), but these sites must be used to investigate long records of volcanism. This issue will cause variability in measured eruption rates which should average out over longer time periods.

Third, there will be a significant bias in the record of global explosive volcanism in Antarctica as a consequence of source locations and magnitude. Northern Hemisphere extratropical eruptions (>23°N) will be under-represented and biased towards very large eruptions with the likelihood of sulfate aerosol moving into the Southern Hemisphere being a function of source latitude and magnitude (Marshall et al., 2021). Thus the Antarctic volcanic record will be biased by larger depositional fluxes for tropical and especially Southern Hemisphere extratropical sources while exhibiting smaller depositional fluxes or even missing events for Northern Hemisphere extratropical eruptions.





The bias in source locations with extratropical Northern Hemisphere eruptions being underrepresented may result in net under-
recording. The estimates of Sigl et al. (Sigl et al., 2015) suggest that around 80% of eruptions giving the greatest global aerosol
loadings are recorded in Antarctica.  Our Antarctic ice core data therefore might underestimate the number of eruptions in each
size class by perhaps 20%, somewhat compensated by the ~20% of eruptions from regional volcanoes recorded in Antarctica
that are tropospheric (Gautier et al., 2019). To first order, both the over-recording due to tropospheric eruptions and the under-
recording of extratropical northern hemisphere eruptions should operate in a similar way through time (excluding any effects
of changing transport strength discussed above) and across a range of eruption magnitudes. Thus the shape of the plots of
number of eruptions versus time and of number against sulfur deposition should be unaffected.
Finally the ice core record in Antarctica is a record of large silicic explosive eruptions (likely mostly M > 6.5). A spatial
analysis of the LaMEVE database of Quaternary explosive eruptions (Fig. 1 in Brown et al., 2014) show that the sources of
these big eruptions are largely in low and mid latitudes. This spatial bias is a consequence of present day plate boundary
distributions. Tectonic settings conducive to forming large silicic magma reservoirs and characterised by caldera forming large
magnitude explosive eruptions are typically in low and mid latitudes where deglaciation effects on melt generation are likely
to be absent or greatly reduced. There are no known M>7 Quaternary eruption from a northern high latitude volcano (>60°N)
(Brown et al., 2014). Thus our findings are not necessarily in contradiction to previous findings (such as Huybers and
Langmuir, 2009) because the Antarctic record is biased towards silicic explosive eruption in the southern hemisphere and
tropical regions.

## 6.   Conclusions

In this study we have extended the study of explosive eruptions recorded at the EDC ice core to 200 ka BP, using a method
that should consistently record large volcanic events through time.  The record mainly represents large magnitude explosive
eruptions with magnitudes of 6.5 or above. We find no systematic variability through time, though there could be a small effect
of transport efficiency manifested in an apparent 20 kyr period that needs to be confirmed in other cores.  There is no sign of
any increase in eruption frequency at deglaciations. This does not of course negate the likelihood that unloading of ice did
cause increased frequencies in regions susceptible to such effects, such as Iceland. However, taking into account the S isotope
evidence that most large eruptions recorded at EDC are stratospheric, our record is probably representative for the major events
influencing climate through stratospheric sulfate. We cannot rule out an effect from volcanism on the balance of $CO_2$
production and removal at deglaciation (Huybers and Langmuir, 2009), but it would have to operate only through smaller high
latitude eruptions and/or submarine volcanism.  Finally we comment that it is difficult to study volcanism in ice cores over a
period longer than 200 ka until the post-depositional effects leading to artefact peaks are better understood.
**Data availability**





The sulfate data on which this paper is based are available at the NCEI paleoclimate data center, at https://www.ncdc.noaa.gov/paleo-search/study/31332; the depths and ages of large volcanic peaks in the EDC ice core that form the basis for Figures 5-7 are listed at Wolff, Eric W; Severi, Mirko (2021): Fluxes of largest volcanic peaks in the EDC sulfate record. PANGAEA, https://doi.org/10.1594/PANGAEA.926087

The age model data, including accumulation rate and thinning factor, is available in the supplement to the Bazin et al. (2013) and in the Pangaea database at https://doi.org/10.1594/PANGAEA.824865

The code used to identify, sum and count peaks, as well as the input data file, is attached as a supplement to this paper.

**Author contribution**

EW, AB and RSJS conceived the idea for this paper. MS provided the sulfate data that were analysed by the Firenze laboratory. EW and SHM studied the ice core data to determine which samples should undergo S isotope analysis. SHM, EAD and LC prepared the samples for S isotope analysis, while AB, HMI and LC carried out those analyses. EWW developed and implemented the sulfate peak identification method, analysed and sensitivity tested the data. AB and JWBR investigated the spectral properties of the data. RSJS and SHM advised about the nature of the volcanic record, including marine and terrestrial data. RSJS, EWW and AB prepared sections of text and all authors edited the text.

**Competing interests**

One of the authors is a member of the CP editorial board. The authors declare that they have no other conflicts of interest.

**Acknowledgments**

This project has been supported by the Leverhulme Trust (RPG-2015-246), by a Royal Society Professorship to EWW, and by a Marie Curie Career Integration Grant to AB. This work is a contribution to the European Project for Ice Coring in Antarctica (EPICA), a joint European Science Foundation/European Commission (EC) scientific programme, funded by the EU and by national contributions from Belgium, Denmark, France, Germany, Italy, The Netherlands, Norway, Sweden, Switzerland and the UK. The main logistic support at Dome C was provided by IPEV and PNRA. We thank Michael Sigl for help with data on estimated emissions of $SO_2$.

**Financial support**

This research has been supported by the Leverhulme Trust (grant RPG-2015-246), by a Royal Society Professorship (grant no. RP/R/180003), and by a Marie Curie Career Integration Grant (CIG14-631752).

**Figure captions**

Figure 1. Comparison of FIC and IC data for two sections of the EDC ice core. In the section from 130 to 150 m, FIC peaks are consistently lower than those of IC while in the section from 390 metres, the concentrations are the same in the two methods.



Figure 2. Examples of volcanic peaks at different depths and ages. The 1257 peak (top panel) is shown after application of the correction described above. The black horizontal bar on each plot represents 5 years at each depth. Dots represent the mid-depths of individual samples.

Figure 3. An example of four artefact peaks in ice aged just over 400 ka.

Figure 4. Distribution of positive and negative peaks exceeding different deposition fluxes, summed over the past 200 kyr.

Figure 5. The deposition flux of sulfate for events with deposition more than 20 mg m$^{-2}$ over the last 200 kyr.

Figure 6: Cumulative eruption numbers over the past 200 kyr recorded in the Dome C ice core for different eruption deposition fluxes of sulfate. Periods of deglaciation marked with blue bar, interglacials with orange bar.

Figure 7: Eruption numbers per millennium (red) and 5 kyr running mean (blue). Periods of deglaciation marked with blue bar, interglacials with orange bar.

Fig. 8. Values of $\Delta^{33}S$ for 21 large volcanic eruptions recorded at Dome C in the last 100 kyr. Values outside a range of $\pm 0.14$‰ are considered to indicate a stratospheric eruption.

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
