# Peer review of "Frequency of large volcanic eruptions over the past 200,000 years"

_Climate of the Past, 2022_

## Referee Comment (RC2)

Review of Wolff et al. "Frequency of large volcanic eruptions over the past 200,000 years"

*Climate of the Past*

11 October 2022

The authors present a new analysis of the EPICA Dome C (EDC) sulfate record, and add new sulfur isotope data, to provide a critical assessment of volcanic eruption frequency over the past 200,000 years. Despite the obvious importance of such an analysis to our understanding of the climate forcing implications of medium- and large-scale volcanic eruptions, and any possible climate-volcanic feedbacks, analyses of volcanic frequency over long timescales are spare. There are good reasons for this, as the authors note in the introduction. The EDC record is well-suited to address the problem, and the authors have a long role in producing and interpreting sulfate data from EDC. The base set of parameters that guide their analysis might be debated, but nonetheless this is a valuable contribution and certainly relevant for *Climate of the Past*. My comments are mainly directed at providing better clarity and insight for the reader, which I hope the authors will consider during revisions.

Line 48: "*Antarctica has rather few local eruptions..*" – that is strongly location-dependent. Probably true for EDC, but certainly not for coastal regions. Please clarify the sentence.

Line 64: "*Antarctic ice cores will record some eruptions which did not reach the stratosphere but are smaller eruptions of more regional origin*" – As written, this seems somewhat inconsistent with the text in line 48. Please provide some clarity here – all Antarctic ice cores, some Antarctic ice cores, and if so which ones, are more or less affected by regional eruptions?

Line 65" "*These can in principle be filtered using sulfur isotope analysis..*" – I agree in principle, but not in practice – at least not yet as far as I understand – we cannot hope to distinguish every eruption with S isotopes. Please add some wording to define the boundaries as they currently stand.

Line 70: "*good depth resolution*" – please be more specific – what constitutes good resolution in this core and time interval of interest?

Lines 91-92: An expanded description of the author's goals, and contribution of the study, would be helpful here. For instance, are they planning to providing any regional or larger context through comparison with other studies? How do the sulfur isotope factor in, etc.? The two lines of course are accurate, but limited in providing the reader with a broader perspective of what the author's hope to do here.

Line 93: I think this section should be titled "Existing EDC sulfate data" or something like that, to clearly distinguish from new methods and data that are being used and contributed here in the "methods" section.

Lines 123-127 and Fig. 3: Without additional analyses, interpretation, or discussion, this section and figure don't add anything to the paper. I would just cite Traversi et al., 2019 and say the sulfate data cannot be confidently interpreted below 200kyr yet.

Line 166: "*This width is justified because diffusion more or less keeps pace with thinning at EDC*" – the phrase "more or less" is a bit unsatisfying, given the stakes of this calculation. I admire the author's ability to intuitively make that judgement, but some quantification justifying the decision would be helpful. If the following sensitivity studies do the job, then just say so.

Lines 187-188: "*..across 21 volcanic sulfate188 events from Dome C between 10.1 and 96.1 ka."* – how and why were these events chosen? Are they representative of sulfate peak size, duration, etc. i.e., what was the sampling strategy and what implications might it have for interpretation (if any)?

Line 198: based on comments above, I think this section should more accurately be titled "..on the East Antarctic plateau" or something similar.

Line 229: "*largest recorded eruption in the timeframe that could accommodate the Toba eruption*"- I'm not exactly sure what that means – that the event with a flux of 133 mgm-2 could be Toba but you're not sure? Can you please clarify the wording and intent.

Lines 259-260: "*There will certainly also be an associated effect on the efficiency of the Brewer Dobson circulation that transports aerosol to the poles through the stratosphere..*" This needs more explanation. What is the specific mechanism here, and what raises your suspicion that it might be in play if it is not present in models? Not saying that models should guide your thinking completely, but as written it's not clear what might be going on.

Line 266: Discussion – I am surprised to get the Discussion section without first seeing the S isotope data presented along with some analysis of error, data characteristics, etc. I think this is an organizational problem of the manuscript which should be corrected, before interpretations and comparisons are made in the Discussion section. Basically, a good portion of this paragraph (lines 267-277) should be in the "methods" section.

Lines 267-277: I don't doubt, based on previous S isotope data and the new data presented here, that most of the sulfate getting to Dome C is stratospheric. But most is not all, and the proportions estimated by the various studies (49 of 64, or 76%, in Gautier; 18 of 21, or 86% here) are different. Is this difference significant? Why or why not? And what implications does that have for interpreting the data solely as a non-regional record (i.e., comments in the introduction)? This is discussed briefly in lines 294-295, but it would help to address it first here I think.

Lines 319-320: True, but given my comment above I would remove reference to this and leave it for a dedicated study.

---

## Author Comment (AC1)

Review in normal type, response in *italic*:

The manuscript presents a 200 ka long volcanic record based on a detailed sulfate record from the Antarctic EDC ice core. To my knowledge, it is the first time a continuous volcanic ice-core record of this duration and quality is being published. Despite being based on a single ice core, the record is very valuable, as it provides a homogeneous, long-term record of major global volcanism. The authors are convincingly demonstrating from sulfur isotopic measurements that the majority of the large sulfate spikes in the record have been injected into the stratosphere and thus have a global impact. Furthermore, the peak shape of the sulfate spikes in the ice core record is fairly 'well preserved' as the diffusion of the sulfate ions in the ice is almost compensated by annual layer thickness thinning. The record is most valuable for better estimating the frequency and magnitude of past volcanism as well as for better assessing the likelihood of major volcanic eruptions in the future. The paper is well written, well referenced and clearly illustrated with figures.

I have just a few suggestion for the authors to consider:

The authors do a comparison to well-known, recent, low-latitude volcanic eruptions to make an estimate of the VEI-sulfate deposition relationship (Table 1). Then they move on to discuss the dependence of latitude of the eruptions for the magnitude of the sulfate deposition in Antarctica/EDC. For reference, it may also be relevant to provide an example of a well-known NH high-latitude eruption such as the Okmok 44 BC eruption, if it has an imprint I EDC? Alternatively, a large Icelandic eruption? Or a statement that none of the well-known NH high-latitude eruption can be detected in the EDC record. An example of the imprint of a 'local' Antarctic eruption would also be illustrative. What about a large eruption from Mt Berlin, Mount Moulton, or Mount Takahe? In particular, if one of the larger peaks in the EDC record were related to local volcanism, it would be good to mention, as an analogy to the Icelandic volcanic imprint in Greenland?

*Thank you for this suggestion. The intention of the table was only to give an approximate idea of how sulfate deposited at Dome C scales with size (in terms of magnitude of eruption and estimated emissions) for the tropical eruptions that probably dominate the record. As you suggest, it can already be seen from Sigl (2013) that the eruptions that show up very strongly in Greenland but that originate from high latitude (such as Laki) are not identified above background in Antarctica. The eruption at 44 BCE, recently identified by McConnell et al (2020) as Okmok, is identified in Antarctica by Sigl (2013), but as having a relatively small deposition flux to Antarctica. We will add paragraph to the text to give this context. Proposed added text:*

*"It was already noted (Sigl et al., 2013; Sigl et al., 2015) that most Icelandic eruptions (such as Laki in 1783 CE and Eldgja in 939 CE) that give large depositions in Greenland cannot be identified in Antarctic cores. Furthermore, these eruptions are estimated to be below the magnitude we would associate with depositions above our chosen threshold of 20 mg m$^{-2}$. An eruption at 44 BCE, which is prominent in Greenland records, was recently identified with the Alaskan Okmok eruption (McConnell et al., 2020), which has been assigned a magnitude of 6.7. The most likely candidate for this eruption in our EDC record has a deposition of 15 mg m$^{-2}$, identical to the value previously noted for this eruption in Antarctica (Sigl et al., 2015). It is therefore likely that for eruptions at high northern latitudes our threshold is closer to M>7."*

*In the case of Antarctic volcanoes the only case we are aware of where an identified Antarctic volcano has been identified in the chemistry (as opposed to the tephra) record is the 17.7 ka extended period of volcanism identified as Takahe by McConnell et al (2017). While this eruption has been identified in the fluoride record from EPICA Dome C, none of the sulfate peaks within the period of high fluoride exceed our threshold. We do not think it helps the narrative to discuss this in the text.*

It is quite remarkable, that one of the largest known volcanic eruptions of the last glacial cycle, the Oruanui, Taupo, eruption occurring close to 25.5 ka, is not pronounced in Fig. 5. This eruption that is

identified with tephra in the WD ice core (Dunbar et al., 2017) and that is associated with very large sulfate deposition in both WD and EDML is classified as a VEI-8 eruption. How come that this very large SH eruption only leaves a weak sulfate imprint in EDC? Likewise, the largest spike in Fig. 6 occurring around 45 ka is much less pronounced in both the EDML and WD ice cores (Lin et al., 2022) questioning its significance.

*Yes, you are emphasising the point that we also make, that the deposition at a single site for a single eruption is not a reliable indicator of the sulfate loading. This point was well-made in the spatial studies carried out at Dome C (e.g. Gautier et al., 2016), and indeed is shown for Oranui in the supplementary tables of Lin et al (2022) who cite a value of only 7 kg km$^{-2}$ for Oranui at EDC (though there is a peak just above 20 kg km$^{-2}$ within 50 cm of that which might more likely be the Oranui peak in EDC).*

With this in mind, the question is how representative the EDC sulfate record is in terms of quantifying global volcanism. For example, I am not convinced that we based on the EDC sulfate record alone can conclude that the Toba 74 ka eruption was not (among) the most significant volcanic climate forcing events of the investigated period, just because it does not show up among the largest spikes in this record. As the authors mention, the sulfate signal of individual eruptions in a single core is subject to great uncertainty.

*You are right to question the level of certainty in our statement about Toba. Proposed revised text:*

*"This raises questions as to whether, in terms of global dispersion of sulfate aerosol, Toba was the most significant volcanic climate forcing event of the past 200 kyr. However additional data from other deep ice cores covering this time period are needed to determine this more certainly."*

Clearly, the authors have no direct way to work around this issue, but it illustrates the need to obtain multiple long-term volcanic records from Antarctica. The Dome Fuji ice core or the Vostok ice core should be good candidates for providing additional information about large volcanic eruptions on this time scale. Could also be that the EDC sulfur isotopic results could provide some additional information?

*The S isotope data for these events has of course been explored in some detail by Crick et al (2021). There is no clear way to derive magnitude of S emissions from those data at present however.*

Figure 8 is very interesting. It is good to know that the majority of the large sulfate spikes we see in the EDC sulfate record are associated with large global/stratospheric volcanic eruptions. We are, however, not being provided with much interpretation of the D33S parameter, except that it is a stratospheric injection indicator. Does it mean anything if the parameter is positive or negative? Does the amplitude of the signal have any significance? There seems to be a few extreme values at around 74 ka and 80 ka. Are those related to specific events? I hope we will learn more are about the interpretation of this dataset, if not in the present MS then in a future publication?

*Indeed these data will be discussed in more detail in a paper in preparation. However we agree we have been too hasty in our description of the S isotope data, so we will add this proposed text:*

*"Mass-independent fractionation of S occurs when sulfur dioxide is oxidised above the ozone layer, producing positive values of $\Delta^{33}S$, followed by (for reasons of mass balance) negative values, so that non-zero values of either sign indicate material that has reached the stratosphere."*

Minor comments:

In the introduction, DEP and ECM are mentioned, but what about the use of liquid conductivity or acidity profiles as indicators of volcanism in ice cores?

Acidity itself is challenging to measure in the ice, so has not been used in a routine way. We agree that in some cores liquid conductivity, though less specific for acid than the solid methods, could be used to identify volcanic peaks but we do not feel that its use is so widespread (or desirable) that we should add it here.

Would it be possible to include the EDC isotope curve in Fig. 5. to make a reference to climate?

*This is a nice suggestion. We will do this. Proposed new caption:*

*"Figure 5. The deposition flux of sulfate for events with deposition more than 20 mg m$^{-2}$ over the last 200 kyr (blue, left axis). Antarctic δD is shown in red (right axis) to indicate the climatic context.."*

*Proposed new figure:*

---

## Author Comment (AC2)

Review in normal type, response in *italic*:

The authors present a new analysis of the EPICA Dome C (EDC) sulfate record, and add new sulfur isotope data, to provide a critical assessment of volcanic eruption frequency over the past 200,000 years. Despite the obvious importance of such an analysis to our understanding of the climate forcing implications of medium- and large-scale volcanic eruptions, and any possible climate-volcanic feedbacks, analyses of volcanic frequency over long timescales are spare. There are good reasons for this, as the authors note in the introduction. The EDC record is well-suited to address the problem, and the authors have a long role in producing and interpreting sulfate data from EDC. The base set of parameters that guide their analysis might be debated, but nonetheless this is a valuable contribution and certainly relevant for Climate of the Past. My comments are mainly directed at providing better clarity and insight for the reader, which I hope the authors will consider during revisions.
*Thank you for this. We appreciate the aim for more clarity.*

Line 48: "Antarctica has rather few local eruptions." – that is strongly location-dependent. Probably true for EDC, but certainly not for coastal regions. Please clarify the sentence.

*We agree that our wording was imprecise. We intended to refer to the frequency of large eruptions leading to a major S loading. We have taken the opportunity to expand our explanation. Proposed wording:*

*"For the region around Antarctica, there are active volcanoes in New Zealand, the Andes, the South Sandwich and South Shetland Islands, and within the continent in Marie Byrd Land and around McMurdo Sound. However the frequency of large eruptions from these areas is expected to be low (notwithstanding an unusual event at the last deglaciation (McConnell et al., 2017)), and the local sources within the continent are far from Dome C. The record of eruptions is therefore likely dominated by those that have reached the stratosphere and have a global climate effect. "*

Line 64: "Antarctic ice cores will record some eruptions which did not reach the stratosphere but are smaller eruptions of more regional origin" – As written, this seems somewhat inconsistent with the text in line 48. Please provide some clarity here – all Antarctic ice cores, some Antarctic ice cores, and if so which ones, are more or less affected by regional eruptions?

*We don't really see an inconsistency here. There will be few local eruptions but still some. We propose a slight revision:*

*"Thirdly, despite the lower frequency of local eruptions, Antarctic ice cores will still record some of these minor eruptions"*

Line 65 "These can in principle be filtered using sulfur isotope analysis.." – I agree in principle, but not in practice – at least not yet as far as I understand – we cannot hope to distinguish every eruption with S isotopes. Please add some wording to define the boundaries as they currently stand.

*I am not sure what the reviewer intends here. In principle every eruption where a significant proportion of the S reaches the stratosphere will undergo mass independent fractionation, and so analysis at high enough depth resolution should identify it. No change planned.*

Line 70: "good depth resolution" – please be more specific – what constitutes good resolution in this core and time interval of interest?

*Resolution that gives at least 5-10 samples across a peak is required. At EDC this then means that better than 5 cm is needed (because of diffusion this applies throughout the 200 kyr period). Proposed text:*

*"resolution (at EDC resolution of better than 5 cm is optimal),"*

Lines 91-92: An expanded description of the author's goals, and contribution of the study, would be helpful here. For instance, are they planning to providing any regional or larger context through comparison with other studies? How do the sulfur isotope factor in, etc.? The two lines of course are accurate, but limited in providing the reader with a broader perspective of what the author's hope to do here.

*OK, we have added a couple of sentences. Proposed additional text:*

*"We use this to assess the variability in recorded eruptions with time and with climate. We discuss how representative the eruption record in Antarctica is, and use sulfur isotope analysis to augment this discussion."*

Line 93: I think this section should be titled "Existing EDC sulfate data" or something like that, to clearly distinguish from new methods and data that are being used and contributed here in the "methods" section.

*While we understand the point, we don't think there is any confusion here for a reader who looks at these sections, and the proposed rewording seems rather convoluted. No change proposed.*

Lines 123-127 and Fig. 3: Without additional analyses, interpretation, or discussion, this section and figure don't add anything to the paper. I would just cite Traversi et al., 2019 and say the sulfate data cannot be confidently interpreted below 200kyr yet.

*It seems to us quite important to explain why we do not use the data below 200 ka. While Traversi et al certainly describe the phenomenon, the examples they show are from deeper than the ones we show in Fig 3 and do not so clearly show why it would be dangerous to include them in an analysis of volcanic signals. We therefore would prefer to leave this section to illustrate the issue to readers, many of whom will be unaware of this phenomenon.*

Line 166: "This width is justified because diffusion more or less keeps pace with thinning at EDC" – the phrase "more or less" is a bit unsatisfying, given the stakes of this calculation. I admire the author's ability to intuitively make that judgement, but some quantification justifying the decision would be helpful. If the following sensitivity studies do the job, then just say so.

*Our wording was not quite right. We now refer to Fig. 2 as visual evidence, and alter the sentence, with a suggested wording:*

*"This suggests that diffusion approximately keeps pace with thinning at EDC."*

Lines 187-188: "..across 21 volcanic sulfate188 events from Dome C between 10.1 and 96.1 ka." – how and why were these events chosen? Are they representative of sulfate peak size, duration, etc. i.e., what was the sampling strategy and what implications might it have for interpretation (if any)?

*We chose a range of sulfate events, including (a) examples of peaks that were strong in both EDC and in the matched peak at Dome Fuji, (b) examples of peaks that were strong at EDC but not in Dome*

*Fuji, and (c) peaks that were smaller near to the strong peaks. They are not therefore strictly representative, but nonetheless they cover a range of ages and size of peak. We suggest an extra sentence:*

*"These included examples of both larger and smaller volcanic peaks in the EDC core."*

Line 198: based on comments above, I think this section should more accurately be titled "..on the East Antarctic plateau" or something similar.

*OK, we have inserted "the EDC core" before Antarctica to be completely clear.*

Line 229: "largest recorded eruption in the timeframe that could accommodate the Toba eruption"- I'm not exactly sure what that means – that the event with a flux of 133 mgm-2 could be Toba but you're not sure? Can you please clarify the wording and intent.

*We have altered the wording here in response to a comment by the other reviewer and hope this also clarifies our intent for this reviewer.*

Lines 259-260: "There will certainly also be an associated effect on the efficiency of the Brewer Dobson circulation that transports aerosol to the poles through the stratosphere.." This needs more explanation. What is the specific mechanism here, and what raises your suspicion that it might be in play if it is not present in models? Not saying that models should guide your thinking completely, but as written it's not clear what might be going on.

*As we have said, we are not aware of any studies that have commented on the strength of the Brewer Dobson circulation in relation to precession (the study we cite by Fu et al looks at the LGM, with a similar stage in the precessional cycle to the present). However, if the ITCZ moves, as Singarayer et al find, then this must affect the B-D circulation. It is not that models don't show it, just that no studies have investigated it as far as we know. No change proposed.*

Line 266: Discussion – I am surprised to get the Discussion section without first seeing the S isotope data presented along with some analysis of error, data characteristics, etc. I think this is an organizational problem of the manuscript which should be corrected, before interpretations and comparisons are made in the Discussion section. Basically, a good portion of this paragraph (lines 267-277) should be in the "methods" section.

*This is indeed a matter of organisation. We saw this study as being primarily about frequencies of recorded eruptions. The use of S isotopes is only introduced here as support for the notion that most eruptions seen in Antarctica are stratospheric, as indicated by the systematic study of the recent past by Gautier. We then felt it would be useful to add our observations of the longer time period, but as they will be discussed in detail in a separate paper (in preparation) it did not seem appropriate to make a separate section for them. Of course we can move a section from the discussion if the editor requires it but we believe this is a matter of taste and prefer to leave as is.*

Lines 267-277: I don't doubt, based on previous S isotope data and the new data presented here, that most of the sulfate getting to Dome C is stratospheric. But most is not all, and the proportions estimated by the various studies (49 of 64, or 76%, in Gautier; 18 of 21, or 86% here) are different. Is this difference significant? Why or why not? And what implications does that have for interpreting the data solely as a non-regional record (i.e., comments in the introduction)? This is discussed briefly in lines 294-295, but it would help to address it first here I think.

*The differences in proportions most likely relate to the size of peaks considered: as we noted all the largest peaks examined by Gautier were stratospheric. Given the (as pointed out by the reviewer) unsystematic way our examples were chosen we do not feel a more quantitative comparison is justified.*

Lines 319-320: True, but given my comment above I would remove reference to this and leave it for a dedicated study.

*We cannot really see a justification for removing a cautionary note such as this which the reviewer agrees is true.*